# Integrative Analyses of Multilevel Omics Reveal Preneoplastic Breast to Possess a Molecular Landscape That is Globally Shared with Invasive Basal-Like Breast Cancer (Running Title: Molecular Landscape of Basal-Like Breast Cancer Progression)

**DOI:** 10.3390/cancers12030722

**Published:** 2020-03-19

**Authors:** Zhenlin Ju, Anjana Bhardwaj, Matthew D. Embury, Harpreet Singh, Preethi H. Gunaratne, Isabelle Bedrosian, Jing Wang

**Affiliations:** 1Department of Bioinformatics and Computational Biology, Houston, TX 77030, USA; zju@mdanderson.org; 2Department of Breast Surgical Oncology, The University of Texas MD Anderson Cancer Center, Houston, TX 77030, USA; abhardwaj@mdanderson.org (A.B.); MDEmbury@mdanderson.org (M.D.E.); s.harpreet20@gmail.com (H.S.); 3Department of Biology and Biochemistry, University of Houston, Houston, TX 77030, USA; phgunara@Central.UH.EDU

**Keywords:** basal-like breast cancer, preneoplastic, miRNAomics, RNA sequencing, integromics

## Abstract

To characterize molecular changes accompanying the stepwise progression to breast cancer and to identify functional target pathways, we performed miRNA and RNA sequencing using MCF10A cell lines based model system that replicates the multi-step progression involving normal, preneoplastic, ductal carcinoma in situ, and invasive carcinoma cells, where the carcinoma most resemble the basal-like subgroup of human breast cancers. These analyses suggest that 70% of miRNA alterations occurred during the initial progression from normal to a preneoplastic stage. Most of these early changes reflected a global upregulation of miRNAs. This was consistent with a global increase in the miRNA-processing enzyme DICER, which was upregulated as a direct result of loss of miRNA let-7b-5p. Several oncogenic and tumor suppressor pathways were also found to change early, prior to histologic stigmata of cancer. Our finding that most genomic changes in the progression to basal-like breast cancer occurred in the earliest stages of histologic progression has implications for breast cancer prevention and selection of appropriate control tissues in molecular studies. Furthermore, in support of a functional significance of let-7b-5p loss, we found its low levels to predict poor disease-free survival and overall survival in breast cancer patients.

## 1. Introduction

According to a well-established model of cancer development, sporadic breast cancer develops through progressive accrual of driver molecular aberrations in the normal breast tissue [1]. On the basis of this model, one would envision early, high-risk lesions to possess a limited amount of molecular alterations, with lesions further down the histologic spectrum (such as ductal carcinoma in situ [DCIS]) showing disproportionally higher rates of mutational change. In contrast to this prevailing notion, several recent studies have found a plethora of molecular changes, such as loss of heterozygosity, miRNA, and gene aberrations in normal tissues immediately adjacent to areas of breast carcinoma [2,3,4,5]. This suggests that a substantial amount of molecular alteration may be occurring early in the tumorigenic process. 

To evaluate these two competing models of the molecular evolution across the preneoplastic state, we seek to comprehensively characterize the genomic changes within a model system that has been shown to closely resemble the development of basal-like human breast cancers [6,7,8,9], an aggressive breast cancer subtype. This model comprises cell lines that recapitulate the key histologic lesions in tumorigenesis: atypia, DCIS, and invasive cancer. Our objectives are to understand the breadth and timing of molecular alterations and their relationship to well-accepted stages of histologic progression and to identify the functional networks that may suggest opportunities for targeted prevention. We performed miRNAomic- and transcriptomic- studies on these samples and integrated the findings across these datasets to identify miRNA-mediated functional gene targets and signaling pathways that underlie both the early and late stages of tumor evolution. Our findings support the premise that a substantial amount of molecular alterations occur before any histologic evidence of cancer, with a global upregulation of miRNA that subsequently drives substantial downstream change in gene expression and pathway regulation. Further, we report a key role for let-7-mediated upregulation of the enzyme DICER in driving the global upregulation in mature miRNAs that we observed during breast cancer progression. let-7 family members are well-known tumor suppressor miRNAs that have reduced expression in cancers [10]. DICER is known to be deregulated in several cancers, but its pattern of deregulation does not appear to conform to those of typical tumor suppressors or oncogenes. Its partial suppression is known to increase tumorigenesis, and complete deletion is required for the inhibition of tumor growth [11]. Its deregulation has also been shown to be variable across cancers. A transient upregulation of DICER is detected in early stages of lung adenocarcinoma, whereas it is downregulated during the most advanced stages of lung cancer [12]. In contrast, DICER expression is shown to be increased in prostate adenocarcinoma and Burkitt’s lymphoma [13]. In the setting of breast cancer, DICER expression varies by subtype, with hormonally sensitive breast cancers expressing higher levels than the hormonally insensitive basal-like subtype of breast cancer [14,15]. Loss of DICER expression is also reported to be associated with breast cancer progression and poor prognosis [14,15].

## 2. Results

### 2.1. An Early Global Increase in miRNAs is Seen during Breast Tumorigenesis

A stepwise approach was taken in order to identify miRNAs that change significantly across the histologic progression model, according to the predefined tumor progression profiles based on the biological hypotheses detailed in Figure 1A. We were particularly interested in identifying miRNAs that followed one of eight expression patterns: early (between MCFA10.P and MCF10.AT1) and continuous increase across the spectrum (G1), early and continuous decrease across the spectrum (G8), early increase followed by a plateau (G2), early decrease followed by a plateau (G7), delayed (between MCF10.AT1 and MCF10.DCIS) increase (G3), delayed decrease (G6), late (between MCF10.DCIS and MCF10.CA1D) increase (G4), and late decrease (G5). 

As the first step, we performed miRNA sequencing (miRNA-seq; Figure 1B). This yielded 1,983 annotated miRNAs. We removed 1,533 miRNAs that were expressed at the same levels across all of the samples (i.e., variance was equal to 0), as these were deemed uninformative for the primary objective of examining the molecular alterations that underlie histologic progression. The remaining 450 miRNAs were used for downstream analysis (Appendix A). In the second step, we applied the ANOVA test to identify miRNAs that were significantly altered across any histologic step along the continuum (normal, preneoplastic, DCIS, and invasive cancer) and used the false discovery rate (FDR) to adjust the type I error due to the multiple comparisons. Lastly, on the basis of the ANOVA results, we employed the Tukey post-hoc pairwise test to identify significant differences. These analyses provided a list of 296 miRNAs that significantly (FDR *p* < 0.05) changed across any point in the histologic spectrum (Figure 2A). Next, K-means clustering was applied to map these 296 miRNAs to the eight predefined groups of expression patterns (Appendix A). Of the 296 significantly altered miRNAs, 213 miRNAs mapped to these groups (Figure 2B,C); 83 miRNAs did not follow any consistent pattern and were excluded from further analyses. We sought to validate the expression pattern of 3 (of 83) random miRNAs in the MCF10A panel by quantitative PCR (qPCR) assay, and all of these miRNAs showed similar expression patterns using both assays (Appendix A). 

Two patterns of interest emerged from the global miRNA analyses. First, most of the miRNAs were upregulated during tumorigenic progression, with expression levels of 177 (83%) of the 213 miRNAs increasing in this multistep model. Second, the majority of the miRNA changes were noted in the initial progression from normal to preneoplastic cells. Using a log fold change of 2 as a threshold and Tukey pairwise comparison, 107 miRNAs were identified to be significantly altered across transition states, of which 80 (75%) (Figure 2B,C) occurred in the transition from normal (MCF10A.P) to preneoplastic (MCF10.AT1), 22 (20%) occurred in the transition from MCF10.AT1 to MCF10.DCIS, and 5 (5%) occurred in the transition from MCF10.DCIS to MCF10.CA1D. Sixty miRNAs were found to be common between the three comparisons (normal to preneoplastic, preneoplastic to DCIS, and DCIS to invasive) (Figure 2D). These common miRNAs are likely to have a role in early-stage, as well as late-stage, transitions in breast cancer progression. Thus, targeting of their downstream genes/pathways may be suitable for prevention, as well as treatment purposes.

### 2.2. let7b-5p Mediated DICER Regulation Causes Early Global miRNA Upregulation 

To study the mechanism underlying the observed global upregulation of mature miRNAs, we tested the hypothesis that an increase in miRNA processing enzyme DICER might be involved. Consistent with this hypothesis, we observed an increase in *DICER* mRNA (1.8 fold, *p* < 0.05, Figure 3A) and DICER protein (2.89 fold, *p* < 0.05, Figure 3B,C) levels in MCF10.AT1 preneoplastic cells, as compared to the normal MCF10A.P cells. miRNA binding prediction databases such as miRTarBase and Target Scan were used to identify miRNA candidates that can directly bind and regulate DICER. miRTarBase predicted six and Target Scan predicted ten miRNAs that overlapped with Next-Gen miRNA sequence expression data in our study. Heatmaps for these putative DICER regulating miRNAs are shown in (Appendix A). Six miRNAs were found to be common between these two prediction algorithms. Out of these six miRNAs- miR-21-5p, miR-195-5p and miR-497-5p failed to show a pattern of expression that is consistent with DICER expression pattern, suggesting that these were unlikely to be regulating DICER in our model system. In contrast, miRNA-29c-3p and let-7b-5p showed a steady down regulation in our breast cancer development model that is consistent with a general upregulation of DICER. Our previous studies have shown miRNA-29c-3p to be a tumor suppressor miRNA that is of relevance to TNBC, and is lost in the MCF10A-based breast cancer progression model consistent with upregulation in DICER [16]. Therefore, we tested the hypothesis that miRNA-29c-3p plays a role in the observed upregulation of DICER1 in the breast cancer progression model. These experiments revealed that although miRNA-29c expression is appropriately lost in the breast cancer progression model [16], and is predicted to regulate DICER, this regulation is not relevant in the setting of breast tumorigenesis (Appendix A). Specifically, these results show that the restoration of miRNA-29c-3p, which is lost in preneoplastic MCF10.AT1 cells by sequence specific miRNA mimics, did not relieve the repression of *DICER* mRNA and DICER protein levels remained unchanged compared to scramble control mimic transfected cells (Appendix A).

let-7 miRNA family members have been previously demonstrated to directly bind and repress DICER1 [17]. We performed several correlative and causal experiments to test if let-7b-5p, a let-7 family member, regulates DICER1 in the setting of breast cancer as well. Although, there is no perfect correlation between let7b-5p expression and DICER expression, there is an overall inverse trend between the molecules that was strongly suggestive of interplay, which we further confirmed through direct assays using let-7b mimics. First, we validated the miRNA sequencing by examining the endogenous expression of let 7b-5p miRNA by qPCR in the progression panel. In these correlative experiments, we found let-7b-5p basal levels decreased by 30% in preneoplastic MCF10.AT1 cells compared to normal MCAF10.P cells (*p* < 0.05), and further decreased by 43% in MCF10.DCIS cells compared to MCF10.AT1 cells (*p* < 0.05; Figure 4A), confirming the miRNA sequencing results. Next, in order to establish the direct association between let-7b-5p and DICER regulation, let-7b-5p was ectopically expressed in preneoplastic MCF10.AT1 cells by transiently transfecting let-7b-5p or scramble control mimics. These experiments revealed an average 42% reduction of endogenous *DICER* mRNA levels (Figure 4B) and an average 60% reduction in DICER protein levels by let-7b-5p mimics, relative to scramble control miRNA (Figure 4C). Lastly, in order to directly assess whether let7b-5p-mediated regulation of DICER is sufficient to account for the global upregulation in miRNAs observed through miRNA-seq, we transfected let-7b-5p or scramble control mimics into MCF10.AT1 cells and tested a panel of 12 randomly selected miRNAs that were noted from the sequencing data to be endogenously increased during breast cancer progression (Figure 2C). Indeed, rescue of let-7b-5p significantly inhibited the expression of a majority of these mature miRNAs (75%, 9 of 12 miRNAs tested; Figure 4D), further validating that let-7b-5p-mediated regulation of DICER was involved in the global miRNA alterations that we observed.

### 2.3. let-7b-5p Predicts Patient Prognosis in TCGA Dataset

In order to investigate the patient relevance of let-7b-5p loss, which we observed during the early steps of breast cancer development, and that has been reported to be associated with higher grade breast tumors in patients [18], we mined The Cancer Genome Atlas (TCGA) breast cancer data to study its association with DSS and OS. The rationale for these studies is based on the logic that an early molecular change may be sufficient to trigger significant changes in downstream molecules (such as DICER change in here, which is sustained later in the breast cancer development) that eventually affects breast cancer patient prognosis. These analyses showed high levels of let-7b-5p to be significantly associated with better DSS and OS (log rank *p* = 0.00126 and *p* = 0.0081) in all breast cancer cases (irrespective of the subtype) (Figure 5). Specifically, we found the median overall survival in patients expressing low levels of let-7b-5p to be 121 months, compared to 212 months in patients expressing high levels of let-7b-5p. For DSS, the median survival in patients with low expression of let-7b-5p was 217 months, and in the group of patients that expressed high levels of let-7b-5p, the DSS was prolonged and never reached the median survival (Figure 5A). 

Within the basal like breast cancer subset, a similar trend was observed with low let-7b-5p associated with poor overall survival, but due to a relatively smaller sample size (n = 179 in basal subtype compared to n = 980 in all subtypes of breast cancer) these trends did not reach statistical significance (Appendix A).

In order to establish if let-7b-5p independently predicts patient survival, we performed a multivariate Cox regression analysis with miRNA let-7b-5p, age, and stage as variables within the model. The results indicate that the low expression of let-7b-5p is significantly associated with poor prognosis of both DSS and OS even after controlling for age and stage of disease (Table 1). Our data are thus consistent with the reported role of let-7b, where its low levels are associated with a higher grade and more aggressive TP53 mutated breast tumors and poorer outcomes [18].

### 2.4. Characterization of Global Gene Expression Changes 

To study the global gene expression, RNA-seq was performed (Figure 1B). Outputs were processed by alignment and assembly of RNA fragments to obtain 17,708 genes; gene expression data were then normalized using the quantile normalization method. Next, we performed variation analysis to remove the low-expression noise genes. Briefly, for each gene, we calculated the standard deviation (SD) of expression levels across samples that were run in triplicate and used at least the 25% quantile of the SDs as cutoffs to select genes. This data-cleaning procedure allowed us to select 13,281 genes that had reliable expression data for downstream analysis. In the second step, we applied an ANOVA to the RNA-seq data and used an FDR of q < 0.05 as a cutoff to identify the significantly changing genes. This analysis allowed us to identify 10,558 genes that were significantly deregulated in at least one of the progression transitions (Appendix A). Next, we applied the Tukey post-hoc pairwise test to identify significant differences. Using a log fold change of at least 2 and Tukey’s HSD (honestly significant difference) adjusted p value of 0.01 as cutoffs, we found that a majority of 6,497 genes (61%) were significantly deregulated in the earliest transition of normal MCF10.P to MCF10.AT1 cells, consistent with the early deregulation of miRNA. A total of 4,022 genes, constituting 38% of the genes, changed in the transition of MCF10.AT1 to MCF10.DCIS cells, and the remaining one fourth of the genes (2,604 genes, 25%) were altered in the transition of MCF10.DCIS to invasive MCF10.CA1D cells (Appendix A). A total of 1,102 genes significantly changed during all three comparisons. We used qPCR as a second method of testing gene expression analysis for 19 randomly selected genes from this list, and for 18 we were able to validate the RNA-seq results (Appendix A). We have previously validated an additional eight genes [16,19]. 

### 2.5. Integration of Multi-Level Omics Data Reveals Regulatory Networks that Define the Progression of Basal-Like Breast Cancer 

#### 2.5.1. Identification of Target Genes

To integrate these gene-level changes with upstream miRNA alterations, a two-step approach was used (as described in Figure 1B). First, the genes were mapped into eight groups based on expression pattern, similar to the groups defined for the miRNA characterization, except reversed. This is because gene expression is expected to increase correspondingly to a miRNA decrease. Thus, genes mapped in the G1 group represented a continuous decrease across the spectrum, in G8 a continuous increase, and so on (Figure 1C). Again, K-means clustering was used to profile the significantly changing genes (10,558) (Appendix A). Of these 10,558 genes, 5,401 genes mapped to the eight expression patterns of interest, specifically, 946 genes in group G1, 1,751 genes in group G2, 452 genes in group G3, 166 genes in group G4, 248 genes in group G5, 299 genes in group G6, 1,126 genes in group G7, and 413 genes in group G8. Next, these 5,401 genes were compared to the predicted gene targets for the 213 miRNAs identified through miRNA sequencing. Of the total 5,401 genes from the RNA sequencing that fell into the eight patterns of interest, 1,299 genes were also identified, using TargetScan, as predicted targets for the miRNAs of interest, and these formed our miRNA-mRNA integrated list. The number of genes/miRNAs per group ranged from 1 target gene (in G5) to 483 target genes (in G2) (Figure 6). 

Interestingly, similar to early miRNA changes, most of the target mRNA alterations also started to occur early; expression of more than 96% of genes changed (of which ~50% genes [648 of 1,299] decreased) during the MCF10A.P to MCF10.AT1 transition. In contrast, only ~3% (38 of 1,299) and ~1% (11 of 1,299) of new gene alterations occurred in the transitions from MCF10.AT1 to MCF10.DCIS and MCF10.DCIS to MCF10.CA1D, respectively. 

#### 2.5.2. Identification of Target Pathways

The list of 1,299 integrated miRNAs and mRNAs was then used to identify and integrate target pathways. We first downloaded 3,195 annotated gene sets/pathways from four major collections (C2, C3, C4, and C6) of the MSigDB database that are of well-known relevance to regulatory motifs and cancers. We applied parametric analysis of gene set enrichment analysis (PAGE) to the RNA-seq data to calculate pathway scores, then took a two-pronged approach to identify miRNA-driven target pathways. In brief, as illustrated in Figure 1B, we first used K-means clustering to group the pathways and applied silhouette analysis to study the separation distance between the resulting clusters. Then, we used a silhouette coefficient of 0.5 as a strict cutoff to remove the pathways that were improperly clustered or very close to the decision boundary (Appendix A). As a result, we were able to assign 1,597 pathways into the eight expression patterns of interest for miRNAs. Next, within each pathway group, we cross-referenced the gene members of the pathways with the 1,299 genes that formed the integrated miRNA-mRNA list to see if there were overlapping genes. For 691 pathways, we found overlapping genes between their gene members and the miRNA-mRNA integrated list, and these were selected as the target pathways. This approach allowed us to narrow down target pathways from a long list of pathways and, furthermore, to integrate the genomic changes of mRNA target genes driven by miRNAs and the target pathways associated with the mRNA genes in the network analysis.

The pathway scores of the 691 target pathways show distinctive patterns between normal MCF10A.P cells and other subsequent stages of histologic progression, and two major patterns were observed (Figure 7). First, most of the pathway alterations occurred in the earliest steps in tumorigenesis (> 99%), with few *de novo* alterations noted in the transition from preneoplastic to DCIS and from DCIS to invasive cancer. Second, the majority (63%, 438 of 691) of pathways appeared to be downregulated early in the progression from the normal-like to preneoplastic state. This global downregulation of pathways is consistent with the global upregulation in miRNAs we observed (Figure 2) and the well-established repressive role of miRNAs in gene regulation. Figure 8A,B shows the miRNAs and their corresponding target genes and pathways that were deregulated in the normal-to-preneoplastic transition in the MCF10A breast cancer progression model (Appendix A). Pathways that were deregulated in the untransformed MCF10A.P-to-MCF10.AT1 preneoplastic transition include EGF, PDGF, MAPK, the JAK/STAT pathway, the IGF-mTOR pathway, the Myc activation pathway, apoptotic cleavage of cell adhesion proteins, cell cycle checkpoints, deadenylation-dependent mRNA decay, G1/S transition, insulin receptor signaling, and the miRNA-mRNA biogenesis pathway (Appendix A). These pathways have been widely known to play a role in breast cancer, and our analysis now suggests their role very early in the molecular and histologic progression to invasive breast cancer. miRNAs and their target genes and pathways, which were implicated in later stages (preneoplastic [MCF10.AT1] to preinvasive [MCF10.DCIS] and preinvasive [MCF10.DCIS] to invasive [MCF10.CA1D]) in the progression model, are shown in Appendix A and listed in Appendix A. Interestingly, using Tukey’s HSD adjusted p value of 0.01 and log fold change of 2 as cutoffs, we found 64 target pathways significantly deregulated in the transition from untransformed cells (MCF10A.P) to preneoplastic cells (MCF10.AT1), but only three pathways (GGCAGTG_MIR3243P, GNF2_NPM1, and MORF_CCNI) changed in the preneoplastic (MCF10.AT1) to DCIS (MCF10.DCIS) transition, and none of the pathways appeared to be de novo deregulated in the DCIS to invasive (MCF10.CA1D) transition based on these statistical criteria. 

### 2.6. Integromics Identifies Top Functional Networks Underlying Transitions in Breast Cancer Progression

We next sought to create a network model to globally capture the regulation of the functional target pathways via their gene mediators and upstream miRNA regulators, and characterize how these networks may differ during tumorigenic progression. For these comparisons, out of 691 target pathways that contain gene targets of the miRNA studied, we focused on the top two pathways that were most altered (based on fold change and adjusted p values obtained from a linear model using LIMMA package in R) within each of the three key histologic transition points. 

Within the first transition (MCF10A.P to MCF10.AT1), MODULE_83 and REACTOME_TRANSLATION were identified as the top two pathways that were activated (Figure 8C). Interestingly, the gene mediators of both of these pathways are key components of mRNA processing and translation machinery. This is in line with the increased cell growth and proliferation that are the hallmarks of the transition of untransformed cells to atypical hyperplastic cells; such growth would require additional building blocks, and therefore our finding of increased mRNA processing and translation in the very early transition of breast cancer progression is plausible. MODULE_83, containing 320 cancer-relevant genes (284 of which occur in our RNA-seq data set), is composed of 27 gene sets, including nucleic acid binding, nuclease activity, protein biosynthesis, and ribosome gene sets, as described in MSigDB (http://www.broadinstitute.org/gsea/msigdb/cards/MODULE_83). This module has been clinically associated with several cancers, including breast cancer. Four (EIF5A, SRP19, SRSF7, and SRPR) of the 284 genes from this network that were identified as the predicted target genes of the indicated miRNAs (Figure 8C) were upregulated in this first transition in breast cancer progression. Most of their upstream miRNAs (miR-181a-5p, miR-204-5p, miR-424-3p, miR-497-5p, miR-195-5p, and miR-424-5p) are known as tumor suppressor miRNAs and were found to be lost in this first transition, causing an upregulation in the gene targets. The gene target EIF5A is a translation initiation factor that is involved in cell cycle progression and functions as a regulator of TP53-dependent apoptosis [20]. Overexpression of EIF5A is required for expression of TP53 to induce apoptosis [20]. SRP19 is potentially involved in RNA binding [21], and SRSF7 is required for pre-mRNA splicing, promoting the nucleocytoplasmic export of mRNA [22]. SRPR is an endoplasmic reticulum signal recognition particle receptor involved in the targeting and translocation of signal-sequence-tagged secretory and membrane proteins across the endoplasmic reticulum [23]. 

The second top pathway network that was upregulated in the transition of untransformed mammary cells to preneoplastic cells was REACTOME_TRANSLATION, which contains 222 genes; 124 of these genes existed in the RNA-seq dataset, and four of these genes (EIF4E, SRPR, SRP19, and SSR1) were identified as target genes in our analyses (Figure 8C). SRPR and SRP19 are shared gene mediators of the top two pathways, as these also had common miRNA regulators (miR-204-5p, miR-497-5p, miR-195-5p, and miR-424-5p). The other two target genes, SSR1 and EIF4E, were upregulated through downregulation of miR-483-5p (which plays a tumor promoter or tumor suppressor role, based on cellular context [24]). SSR1 is an endoplasmic reticulum membrane receptor that is involved in protein translocation across the endoplasmic reticulum membrane. EIF4E, another gene mediator of the REACTOME_ TRANSLATION network, is an mRNA 5’ cap-binding protein that is involved in translation initiation, and its upregulation is involved in cancer development and progression [25]. 

For the preneoplastic (MCF10.AT1) to DCIS (MCF10.DCIS) transition in breast cancer development, GNF2_NPM1 and GGCAGTG_MIR3243P were the top two pathway networks that were found to be significantly dysregulated (by at least twofold; Figure 8C). GGCAGTG_MIR3243P (genes that have a binding site of miR-324-3p in their 3’UTR) is a pathway that was found to be downregulated through its gene target MEIS1 (Meis homeobox 1) in the preneoplastic-to-DCIS transition. Not much is known about the role of MEIS1 in cancer (Figure 8C). Downregulation of the GNF2_NPM1 module occurs through downregulation of target gene CPSF6 (Figure 8C). CPSF6 (cleavage and polyadenylation specificity factor subunit 6) activates pre-mRNA 3’-end processing and the maturation and export of mRNAs. Interestingly, its upstream regulator, miR-30e-5p, switched from being downregulated in the early transition (MCF10A.P to MCF10.AT1) to upregulated in this mid transition (MCF10.AT1 to MCF10.DCIS); its upregulation could lead to inhibition of CPSF6, a gene modulator of translation, and eventually cause a decrease in translation. This would suggest the context-specific switch in the role of a miRNA and the context-specific preferential use of a gene target to achieve inhibition in translation, an outcome that is required for a given developmental step in cancer. 

Lastly, none of the pathways were found to be significantly dysregulated in the transition from the MCF10.DCIS to invasive (MCF10.CA1D) stage. 

### 2.7. Targeting of Functional Networks Inhibits Cell Proliferation and Colonizing Ability of Preneoplastic Cells

In order to validate the biologic relevance of the miRNA-driven pathways that were predicted to be deregulated in the early transition (MCF10.AP to MCF10.AT1), we performed loss of function studies. Our integromic analyses showed an activation of Module 83 and Reactome translation networks through increased expression of six gene mediators- SRSF7, SRP19, EIF5A, SRPR, SSR1, and EIF4E. Using Mission siRNA, we knocked-down expression of all of these six genes individually and studied their functional role in cell proliferation and colonizing ability of preneoplastic MCF10.AT1 cells. Up to two different siRNA constructs per indicated genes were tested (Assay IDs provided in Appendix A). Depletion of four (SRP19, SRPR, EI5FA, and EIF4E) of the six genes significantly inhibited the cellular proliferation of MCF10.AT1 cells (that ranged from 28% to 64% inhibition) compared to scramble control siRNA (Figure 9A). Depletion of all four of these genes (SRPR, EIF5A, EIF4E, and SRP19) also significantly inhibited the colonizing ability (ranging from 40% to 80% inhibition) of preneoplastic MCF10.AT1 Cells (Figure 9B,C). In order to assure the efficiency of gene knock down in the MTT and colony formation assays, a parallel set of cells were treated with siRNA constructs and qPCR was performed. The amount of average depletion ranged from 2-fold to 17.78-fold (Appendix A) for these transcripts. These proof of principle studies validate the functional relevance of miRNA driven pathway networks that are generated by multilevel integromic studies, thus, their importance as central mediators in breast cancer progression and the importance of their targeting for breast cancer prevention. 

## 3. Discussion

While much of the genomic analysis of breast cancer is focused on the characterization of invasive disease, and more recently of in situ carcinoma, the extent of genomic alterations earlier in the progression spectrum has remained largely unexamined. Our findings demonstrate that the majority of molecular changes leading up to invasive cancer have already occurred prior to any histologic evidence of carcinoma. These results extend the latest reports that genetic aberrations are present in histologically normal tissues surrounding tumor tissue by showing the comprehensive and complex degree of such early molecular change during breast tumorigenesis and support the premise of a change in the molecular field that precedes the development of breast cancer. Further, we note substantial upregulation of the miRNA network, which we show is driven at least in part let7b-5p mediated by upregulation of DICER, and that the resulting downstream changes are functionally relevant, conferring important biologic advantage. 

let-7 is a well-known tumor suppressor miRNA relevant to several cancers, including breast cancer, and it has previously been shown to regulate DICER1 expression through direct binding within coding regions or the 3’UTR [26] and to establish an autoregulatory negative feedback loop to regulate DICER in the context of oral cancer cells [27]. While let- 7, miRNA-103/107 and miRNA-192 have been reported to target DICER [28], in our analyses, miRNAs other than let-7 did not make it to the list of potential targets based on the 5% context score cut off filter applied in our study and were not tested further.

The relevance of let-7b to the biology of breast cancer is further supported by studies that show low levels of let-7b to be associated with higher grade breast tumors, and more aggressive TP53 mutated tumors [18]- factors associated with poorer outcome. Similarly, we found poorer disease-specific survival and overall survival in the TCGA breast cancer dataset in patients with low let-7b. 

Our studies extend the relevance of let-7b to early steps of breast tumorigenesis. We show the presence of a let-7-mediated autoregulatory negative feedback loop resulting in early, sustained loss of let-7b-5p which causes an increase in DICER mRNA and protein, eventually resulting in a global upregulation in miRNAs in preneoplastic cells. Interestingly, in our multi-step progression model, we note that expression of let-7b-5p, while low in the preneoplastic and precancerous states, are restored in later stages representative of invasive cancer within the MCF10A panel-based breast cancer progression model. This finding is supported by clinical data demonstrating heterogeneity of let- 7b expression in breast tumors, with relatively higher levels in a subset of patients. These findings suggest the dynamic nature of molecular change during tumorigenesis, such that early changes may, with additional accumulation of genomic instability, result in heterogenous patterns of expression during later stages of tumorigenesis, as also indicated by the Kaplan-Meier survival curves where low levels of let7b-5p predict worse survival in breast cancer cases. 

Our finding that the majority of miRNA and gene alterations that defined the DCIS (MCF10. DCIS) to invasive (MCF10.CA1D) transition are already present in the initial transition of normal, untransformed cells (MCF10A.P) to preneoplastic cells (MCF10.AT1) in breast cancer builds on similar studies reported in the literature. Using micro-dissected cells from atypical ductal hyperplasia, DCIS, and invasive cancer lesions and matched controls obtained from healthy normal women, Ma et al [29] performed gene expression profiling and reported that a significant number of transcriptional changes are already present in atypical ductal hyperplasia lesions and persist through DCIS and invasive cancer. Similarly, using MMTV-PyMT (a mouse model of luminal B-type breast cancer), Cai et al [30] also found that for genes differentially expressed in the late carcinoma stage, the expression alteration was initiated at the hyperplasia stage. Lastly, in a gene expression profiling study using the MCF10A-based breast cancer progression model, Rhee et al [31] also found about 75% of the gene expression changes defined the untransformed MCF10A.P-to-benign transition and only 25% of gene expression changes occurred in the benign-to-DCIS transition. The present manuscript strengthens the concept of early gene-level changes in the preneoplastic breast. Most importantly, it also significantly adds to the understanding of the molecular basis of basal-like breast cancer progression by revealing the timing and nature of miRNA-, gene-, and pathway-level alterations. By integrating these multilevel omics data, we identified the functional networks of miRNA-driven genes and pathways at each transition in the development of basal-like breast cancer, which further established that most molecular deregulations had its roots in the preneoplastic cells. By using a model system that recapitulates the complete spectrum of breast cancer progression, we believe that we have captured a complete multilevel omics landscape of all the major steps of progression. 

Understanding the kinetics of the molecular deregulations is especially important for identifying effective and novel targets for prevention that are driven from a preneoplastic state, rather than invasive cancer, and also in the selection of appropriate controls for identifying and validating molecular deregulations that are driven from the invasive or DCIS step of breast cancer progression. For such analyses, it is common practice to compare cancer tissues to adjacent normal tissues [32]; however, numerous studies, including a previous study of ours [33], suggest that histologically normal-looking tissue adjacent to tumor tissue is not molecularly normal [2,3,4,5]. Therefore, using preneoplastic/adjacent normal tissue samples would not be an appropriate control for identification of any meaningful molecular changes in DCIS/cancer tissue. One such potential problem is evident from the reports of a global downregulation of miRNAs in cancers, including breast cancer [34], where adjacent normal tissue was used as a control. The reported global downregulation of miRNAs in breast cancer may seem contrary to our findings, where we find an initial global upregulation in mature miRNAs and DICER, the enzyme that is involved in miRNA processing during breast cancer progression. By comprehensively studying the miRNA levels across the entire spectrum of breast cancer progression, our results specifically suggest that there is a global upregulation in DICER and mature miRNAs from untransformed cells to preneoplastic cells (MCF10.AT1) that stays high through DCIS, and that there is an eventual global downregulation of DICER and miRNAs in invasive cancer (MCF10.CA1D) compared to atypia. Therefore, characterizing cancers by using comparisons to atypia samples—or adjacent normal tissue, which also bear many molecular hallmarks of the cancer state—would skew the findings, and in this particular instance, suggest a global miRNA decrease in cancer tissues, rather than the global increase in miRNA shown in our report. 

The early global miRNA deregulations that we observed in our study also correlate with deregulations in gene and pathways. Interestingly, we found a lesser proportion of genes to be downregulated (~50% of genes compared to about 70% of miRNAs being upregulated) during the normal-to-preneoplastic transition, which suggests the complexity of gene regulation involving mechanisms other than miRNAs. This would also suggest that a proportion of miRNAs may even activate expression of genes, a regulation that is occasionally reported in the literature [35]. 

Integrated analysis of this miRNAomics and transcriptomics dataset has allowed us to fully capitalize on the data generated by the multilevel omics platforms in order to identify functionally relevant gene targets of miRNAs and biologically meaningful networks. In particular, integrating these datasets has strengthened our analyses and allowed us to (i) identify functional target genes and pathways (of an miRNA), (ii) exclude any false-positive gene alterations, (iii) focus on biologically relevant pathways where several gene mediators are targets of an miRNA that also changes, and (iv) identify miRNA-driven functional gene and pathway networks that are specific to a given transition in the development of breast cancer. The networks created by this multilevel functional integration revealed the upregulation of mRNA translation in the early progression to a preneoplastic state, which is in line with the well-established role of cell growth and proliferation in the early steps of breast cancer development, clinically represented by hyperplasia and atypia. Once the cell growth and proliferation have occurred, the preneoplastic cells gain the ability to acquire additional characteristics that are needed to become neoplastic. In agreement with this concept, the increased mRNA processing and translation would no longer be needed, and that is exactly what we found in the preneoplastic-to-DCIS transition. Lastly, the fact that no novel miRNA-driven pathway-level changes occurred in the late transition from DCIS to invasive breast cancer suggests the role of microenvironment alterations, particularly the presence of a protumor immune-suppressive microenvironment to drive breast cancer progression from DCIS to invasive breast cancer, as has been suggested [36]. 

## 4. Materials and Methods

### 4.1. MCF10A Model System of Breast Cancer Progression and Cell Transfections

We used the MCF10A cell line-based isogenic cell model system, which comprises the entire range of human breast cancer progression [6,7,8,9]. As described previously [16], MCF10A.P is an untransformed, immortalized human mammary epithelial cell line and does not develop tumors when injected in mice. The MCF10.AT1 cell line was generated by HRAS transformation of MCF10A.P cells and forms atypical preneoplasia in mice [7]. MCF10.DCIS cells form comedo DCIS lesions in immune-deficient mice [37]. The MCF10.CA1D and MCF10.CA1H cell lines were derived from MCF10.AT1 xenografts and form malignant tumors in xenograft mouse models. We purchased the MCF10A.P cell line from American type culture collection (ATCC) and obtained the MCF10.AT1, MCF10.CA1D, and MCF10.CA1H cell lines from Karmanos Cancer Center, Detroit, MI, under a Materials Transfer Agreement. MCF10.DCIS cells were purchased from Wayne State University, Detroit, MI. All the cell lines used in the study were authenticated by the selling agency and used within the first ten passages. We have previously tested [19] and found these cell lines to be estrogen receptor, progesterone receptor, and HER2 negative using immunohistochemistry, consistent with the clinical assays used to define the basal-like subtype of breast cancer. 

To test miRNA let-7b-5p or miRNA-29c-3p’s effect on DICER expression, subconfluently growing MCF10.AT1 cells (about 175,000 cells/ well) that were plated in 6-well dishes were transfected with mirVana let-7b-5p miRNA mimic (MC11050) or random scramble miRNA mimic using Lipofectamine 2000 (ThermoFisher Scientific, Waltham, MA) according to the manufacturer’s instructions. 

### 4.2. RNA Extraction and qPCR

As described previously [16], total RNA, including miRNA, was extracted from cells by using the miRNeasy Mini Kit (Qiagen, Germantown, MD), an extraction method that efficiently preserves the small RNA (10-200 nucleotides) fraction. cDNA was prepared from RNA by using the iScript cDNA Synthesis Kit (Bio-Rad, Hercules, CA). The target mRNA levels were measured with respect to a loading control (ribosomal protein L19) by using the SYBR green-based qPCR method, as described previously [19]. The primer sequences used for all gene targets tested are in Appendix A. Levels of mature miRNAs were quantified by qPCR using the TaqMan miRNA assay (ThermoFisher Scientific) following the manufacturer’s instructions.

### 4.3. RNA Sequencing and Data Analysis

Total RNA was also processed for RNA sequencing using the HiSeq 2000 platform, as described earlier [16]. RNA sequencing yielded 30–40 million read pairs for each sample. The data were first mapped to the human genome (UCSC Genome Browser assembly hg19) using TopHat2 [38], then the abundant transcripts were estimated by Cufflinks [39]. Gene expression values were normalized by quantile normalization methods. The significant RNAs were identified by ANOVA and the Tukey post-hoc test, as described above. 

### 4.4. Small RNA Sequencing

Integrity of the extracted total RNA was measured using the Agilent 2100 Bioanalyzer (Santa Clara, CA, https://www.agilent.com/en/product/automated-electrophoresis/bioanalyzer-systems/bioanalyzer-instrument/2100-bioanalyzer-instrument-228250), and only samples that passed strict quality-control standards were processed for next-generation sequencing performed by the Sequencing Core at Baylor College of Medicine, Houston, TX. For small RNA library construction, which yields 25–30 million reads per library, RNA samples were prepared using the DGE-Small RNA Sample Prep Kit (Illumina, San Diego, CA), as described previously, and were analyzed on the Illumina HiSeq 2000 (https://www.illumina.com/documents/products/datasheets/datasheet_hiseq2000.pdf) platform [16].

### 4.5. Analysis of Small RNA Sequencing 

The small RNA-seq data generated in this study was submitted to the GEO database under accession number GSE93740. Curated RNA-seq data are shown in Appendix A. A total of 1,983 miRNAs were annotated, but 1,533 of them were expressed constantly at the same levels across all the samples and thus removed. The remaining 450 mature miRNAs were used for downstream analysis. miRNAs that had significantly increased or decreased expression in any tumor progression step were identified by ANOVA. Furthermore, the Tukey post-hoc pairwise test was used to determine where the significant differences lie. The p values were adjusted for the type I error rate due to the multiple comparisons. A false discovery rate (FDR) of < 0.05 was considered statistically significant. 

### 4.6. Target Gene Prediction

Targets of the significant miRNAs were identified through the database TargetScanHuman (release 7.1; http://www.targetscan.org/vert_71) based on their context score. For miRNAs that had multiple hits, we used the lower 5% quantile of the total context scores as a strict cutoff to choose the most reliable gene targets. As described previously [40], for each predicted target of each miRNA, the sum of the context scores for the miRNA sites was calculated as the total context score. Predicted targets of each miRNA family were sorted by total context score, and the miRNA with the most favorable (lowest) total context score in the family was selected as its representative miRNA.

### 4.7. TCGA Data Mining

We studied the association between let-7b-5p levels and disease-specific survival (DSS) and overall survival (OS) in the breast cancer dataset (TCGA, https://tcgadata.nci.nih.gov/tcga/). All subtypes of primary breast cancer samples (n = 1,013) described within the TCGA consortium were used for this analysis. As described previously, for these analyses, only the DSS and OS outcomes were used to find the correlation with let-7b-5p expression levels. Other clinical or demographic variables were not considered or adjusted for. Briefly, we used the median expression value of let-7b-5p to split the patients into two groups that expressed a high level of let-7b-5p (High) and a low level of let-7b-5p (Low), then performed Kaplan-Meier survival curve analysis and log-rank testing to estimate the significance of DSS and OS differences between the two patient groups. A log-rank test p value of <0.05 was considered as statistically significant.

### 4.8. Integrative Analysis of Biological Pathways

Genomic pathways: To assess genomic pathways, we downloaded 3,195 gene sets/biological pathways from the Molecular Signatures Database (MSigDB, http://software.broadinstitute.org/gsea/msigdb) and applied parametric analysis of gene set enrichment (PAGE) to the RNA-seq gene expression data to calculate pathway scores [41], which are derived from the statistic Z score for a given dataset. To identify target pathways, the K-means clustering algorithm [42] was first used to group miRNAs, mRNAs, and pathways, and interesting groups of molecules and pathways were selected according to the eight predefined tumor progression profiles based on our biological hypothesis. The miRNA target genes of an interesting group were then used to determine the target pathways of that group. Figure 1B represents the workflow to identify the miRNAs and their target genes and pathways that define breast cancer progression. 

### 4.9. Visualization of Gene Expression and Biological Networks

Heat maps with one-way or two-way unsupervised hierarchical clustering analysis were drawn to visualize gene/protein expression patterns. Ward linkage was used as the agglomeration rule and Pearson’s correlation coefficients were used as the dissimilarity metric in the hierarchical clustering analysis. The biological networks between miRNAs, target mRNAs, and pathways were created on the basis of Pearson’s correlation coefficients. 

### 4.10. Western Blotting

Endogenous DICER and vinculin protein levels were measured in subconfluently growing MCF10A panel cells and MCF10.AT1 cells that were transfected with scramble or let-7b-5p/ miRNA-29c-3p mimic by performing Western blotting, as described previously, using DICER antibody (CST# 5362, dilution 1:1000) and vinculin (ab# 130007, 1:2000). 

### 4.11. Functional Assays

Cellular proliferation was measured by MTT assay, as described [19] previously. Briefly, MCF10.AT1 cells (3000–3500 cells/ well) were plated in a 96-well dish and transfected with either negative control siRNA or siRNA of interest at the final concentration of 50 μM using lipofectamine 2000. Forty-eight hours after the transfection, media was replaced with fresh complete media. Twenty-four to forty-eight hours after the media change, MTT reagent was added and incubated with cells for 3 h. The formazan precipitates that were formed in living cells were dissolved in DMSO and the color was measured at 570 nM. 

The ability of single cells to form colonies was measured by plating scramble control/ siRNA of interest transfected MCF10.AT1 cells. The cells were transfected in 6-well dishes that were about 70% confluent (175,000 cells/well) using lipofectamine 2000. Forty-eight hours after the transfection, cells were trypsinized and 70 cells/well were plated in 6-well dishes in complete media. The number of clones (a cluster of more than 50 cells) that were formed nine days later were stained with crystal violet stain and manually counted. 

### 4.12. Statistical Analysis

Both miRNA and RNA-seq experiments were performed in triplicate, allowing for statistical significance analysis by ANOVA and the Tukey post-hoc pairwise test. All bioinformatics and statistical analyses were performed using R 3.5.1 (https://www.r-project.org/) and R packages in Bioconductor (https://www.bioconductor.org/).

## 5. Conclusions

In conclusion, our miRNA-seq, RNA-seq, and integromics analyses suggest that a majority of miRNA-, gene-, and pathway-level deregulation occurs at the very earliest stages in the histologic progression to basal-like breast cancer. Although our integromics analysis is based on a cell model system, the basic principles of early molecular change in the breast parenchyma have also been documented in small studies of patient samples. Collectively, the data emerging from these studies suggest significant opportunities to use the early molecular changes to better understand breast cancer risk and develop novel targeted chemoprevention strategies for breast cancer. 

## Figures and Tables

**Figure 1 cancers-12-00722-f001:**
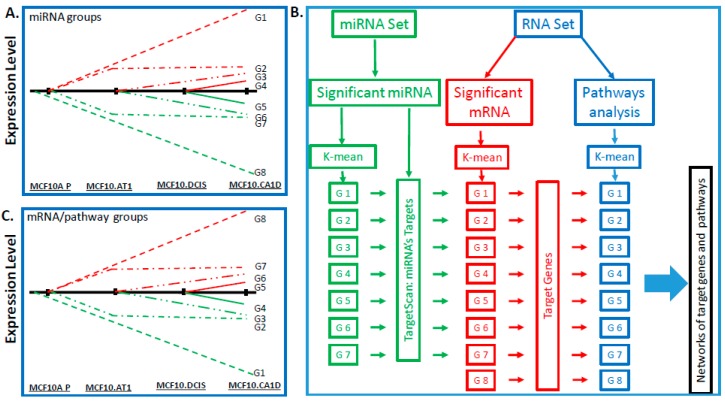
Bioinformatics pipeline. Schematics showing eight patterns of interest (including early changes, early sustained changes, and late changes) that were used as criteria for identifying meaningful changes in (**A**) miRNAs and in (**C**) mRNAs and pathways that define breast cancer progression across a panel of cell lines comprising normal-like (MCF10A.P), preneoplastic (MCF10.AT1), DCIS (MCF10.DCIS), and invasive cancer (MCF10.CA1D) cells. The red lines (Figure 1A,C) indicate that miRNA/mRNA expression levels and pathway scores are higher in progressive stages and the green lines indicate that miRNA/mRNA expression levels and pathway scores are lower in progressive stages compared to MCF10A.P. (**B**) Flowchart showing the bioinformatics pipeline for processing of miRNA- and RNA-sequencing datasets to derive target genes and pathways. First, statistical significance analysis (as described in Methods) was used to identify meaningful miRNAs and RNAs across the breast cancer progression steps, and parametric analysis of gene set enrichment (PAGE) was used to calculate scores of the pathways. Then, a k-mean clustering analysis was used to group the significant miRNAs, RNAs, or pathways for the patterns of interest. TargetScan was applied to determine miRNAs’ targets, which were used to determine the target pathways, if the target genes are among the pathway gene membership. At the last, networks were drawn to visualize the relationships between miRNA, target genes, and pathways. None of the miRNA conforming to the expression pattern of G8 were identified, therefore G8 is missing in the miRNA set.

**Figure 2 cancers-12-00722-f002:**
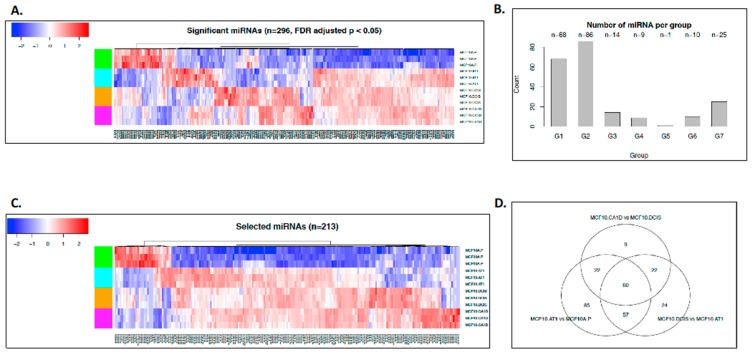
miRNAs that changed significantly during breast cancer progression. (**A**) Heat map showing the expression pattern of all miRNAs (n = 296) that significantly changed across the spectrum of breast cancer progression comprising normal-like (MCF10A.P), preneoplastic (MCF10.AT1), DCIS (MCF10.DCIS), and invasive cancer (MCF10.CA1D) cells. The color key shows blue as the lowest expression and red as the highest expression. (**B**) Bar diagram showing the number of miRNAs that mapped to the predefined groups of interest. (**C**) Heat map showing the expression pattern of 213 miRNAs that mapped to one of the eight predefined groups of interest, which depicts global miRNA changes during breast cancer progression. (**D**) Venn diagram showing the number of miRNA changes that were distinct or shared among normal to preneoplastic (MCF10A.P to MCF10.AT1), preneoplastic to DCIS (MCF10.AT1 to MCF10.DCIS), and DCIS to invasive cancer (MCF10.DCIS to MCF10.CA1D).

**Figure 3 cancers-12-00722-f003:**
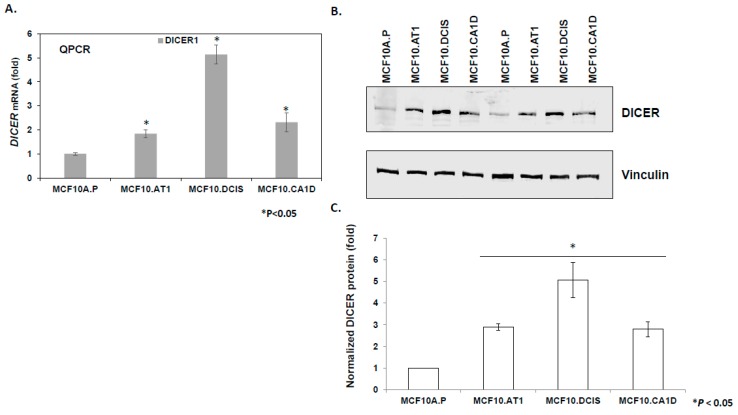
DICER upregulation during breast cancer progression. (A) Relative mean expression of endogenous DICER1 mRNA assayed by qPCR across the MCF10A-based breast cancer progression panel. DICER1 mRNA levels were first normalized to a housekeeping gene, RPL19. The Ct values were converted to fold change using the delta Ct method and setting the lowest gene expression (MCF10A.P cells, in this case) as 1. (B and C) Western blot analysis and its quantification showing upregulation in endogenous DICER1 protein in the breast cancer progression panel. DICER protein levels were normalized to the loading control, vinculin. The relative DICER expression was set as 1 in MCF10A.P cells, and the relative fold change was calculated in other cell lines. The graph represents mean values + SEM. p values were calculated using student’s t test. * indicates *p* < 0.05.

**Figure 4 cancers-12-00722-f004:**
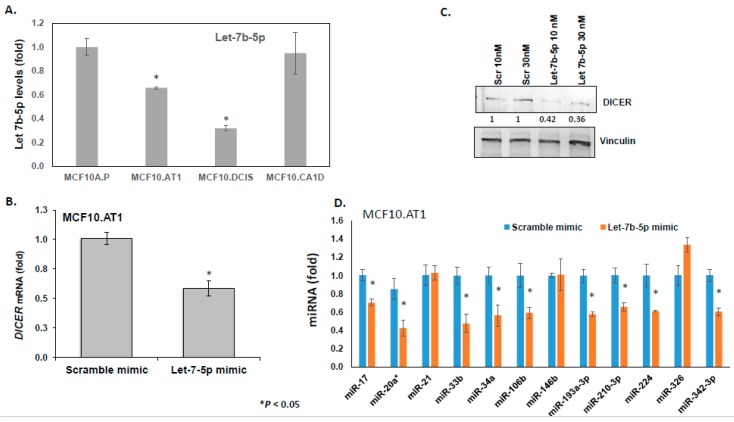
let-7b-mediated DICER regulation causes global miRNA changes. (**A**) Relative mean expression of endogenous let-7b-5p miRNA assayed by TaqMan-based qPCR across the breast cancer progression panel. let-7b levels were first normalized to a small nucleolar RNA, RNU44. The Ct values were converted to fold change using the delta Ct method and setting the lowest gene expression (MCF10A.P cells, in this case) as 1. (**B**) Relative mean expression of endogenous DICER1 mRNA assayed by qPCR in MCF10.AT1 cells transfected with either scramble mimic or let-7b-5p mimics. DICER1 mRNA levels were first normalized to a housekeeping gene, RPL19. The Ct values in let-7b-5p mimic-treated cells were converted to fold change using the delta Ct method and setting the gene expression in scramble-treated MCF10.AT1 cells as 1. (**C**) Western blot analysis and its quantification showing repression in endogenous DICER1 protein caused by let-7b-5p mimics compared to scramble control mimics in preneoplastic MCF10.AT1 cells. DICER protein levels were normalized to a loading control, vinculin. The graphs represent mean values + SEM. (**D**) Relative mean expression of indicated miRNAs as assayed by qPCR in MCF10.AT1 cells transfected with either scramble mimic or let-7b-5p mimics. miRNA levels were first normalized to a loading control, RNU44. The Ct values in let-7b-5p mimic-treated cells were converted to fold change using the delta Ct method and setting the gene expression in scramble-treated MCF10.AT1 cells as 1. The graphs represent mean values + SEM. p values were calculated using student’s t test. * indicates *p* < 0.05.

**Figure 5 cancers-12-00722-f005:**
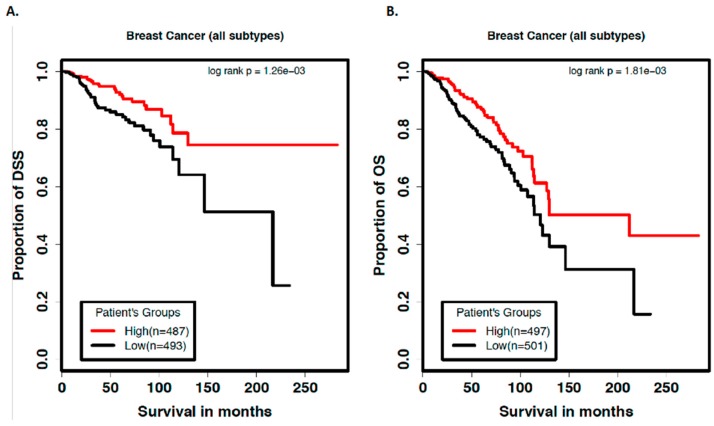
Low levels of let-7b-5p predict poor prognosis in breast cancer patients. Kaplan-Meier curves showing (**A**) disease-specific survival and (**B**) overall survival (in months) in breast cancer patients expressing high vs low levels of let-7b-5p.

**Figure 6 cancers-12-00722-f006:**
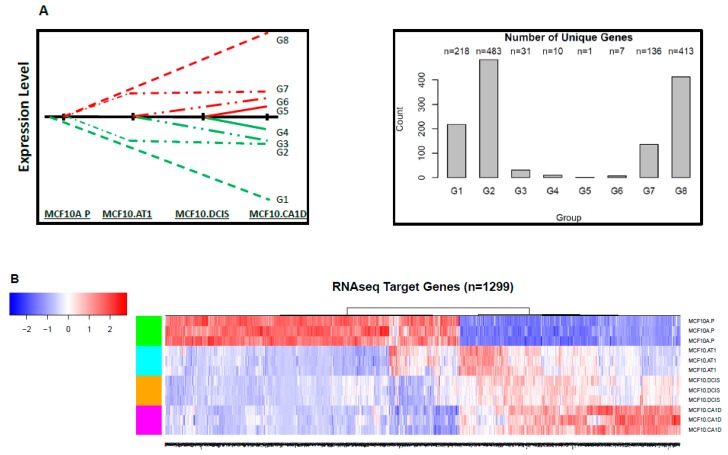
Target genes that changed significantly during breast cancer progression. (**A**) Schematic showing the eight patterns of interest and the number of target genes that mapped to each of these groups. (**B**) Heat map showing the expression pattern of target genes that significantly changed and thus depict global mRNA changes across the spectrum of breast cancer progression. The color key shows blue as the lowest expression and red as the highest expression.

**Figure 7 cancers-12-00722-f007:**
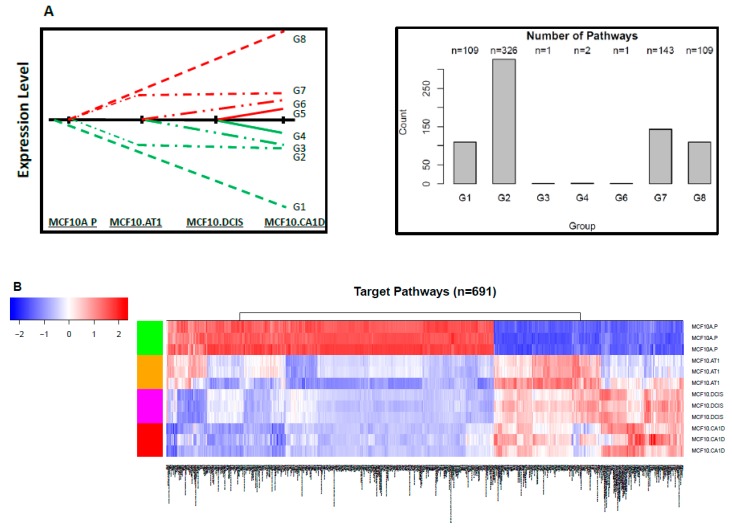
Target pathways that changed significantly during breast cancer progression. (**A**) Schematic showing the eight patterns of interest and the number of target pathways that mapped to each group. (**B**) Heat map showing the expression pattern of target pathways that significantly changed, thus depicting global target pathway changes across the spectrum of breast cancer progression. The color key shows blue as the lowest expression and red as the highest expression.

**Figure 8 cancers-12-00722-f008:**
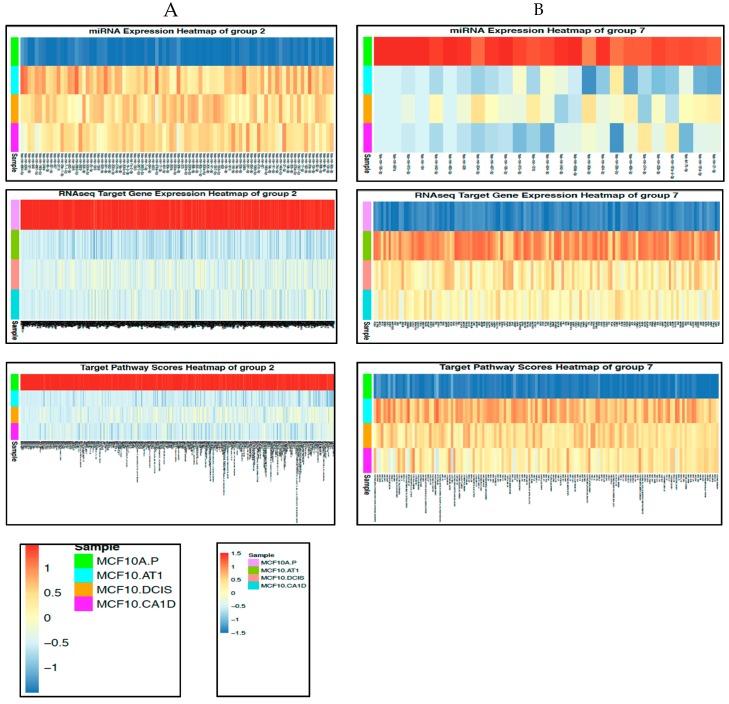
miRNAs and their target genes and target pathways that changed across histological progression. (**A** and **B**) Heat maps showing the miRNAs and their target genes and pathways that mapped to (**A**) Group 2 or (**B**) Group 7 and represent early changes across the preneoplastic transition of breast cancer progression. The color key shows blue as the lowest expression and red as the highest expression. (**C**) Network diagrams for the top two pathways derived from the transitions of normal MCF10A.P to preneoplastic MCF10.AT1 cells and MCF10.AT1 to MCF10.DCIS cells. The large central sphere represents the core pathway, with more peripheral circles representing genes that regulate the pathway. The sizes of these peripheral circles are scaled to show their relative impact on the pathway. Selected miRNAs are shown in triangles, with their gene targets represented within squares. Magenta nodes represent activation of the pathway/gene/miRNA, while blue nodes represent suppressed pathways/genes/miRNAs. Green lines between green nodes represent upregulation; green lines between a green and red node represent repression. Similarly, red lines between red nodes represent upregulation, and red lines between a green and red node represent inhibition.

**Figure 9 cancers-12-00722-f009:**
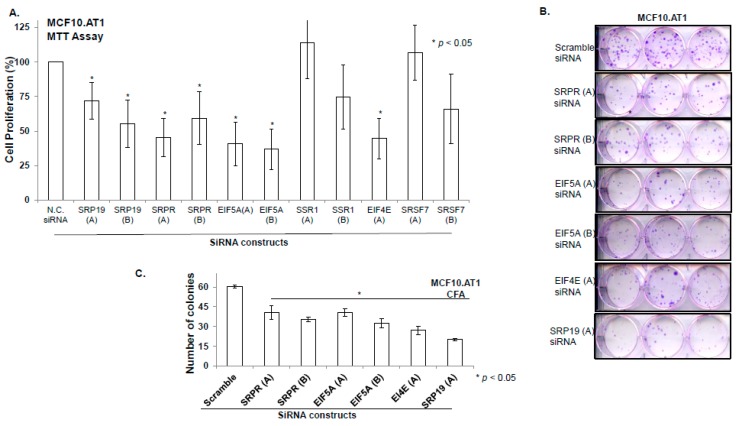
Depletion of key gene mediators of miRNAs driven target pathways/ networks inhibits cellular proliferation and colonizing ability of preneoplastic MCF10.AT1 cells. (**A** and **B**) Bar diagram showing % cell proliferation (**A**) and quantification of colonizing ability (**B**) of MCF10.AT1 cells transfected with either scramble negative control siRNA or siRNA of the indicated genes. The graphs represent mean values + SEM. P values were calculated using students’ unpaired t test. * indicates *p* < 0.05. (**C**) Crystal violet stained colonies formed by MCF10.AT1 cells transfected with siRNAs of indicated genes or scramble negative control siRNA.

**Table 1 cancers-12-00722-t001:** Multivariate Cox model of factors associated with survival outcome.

	Variable		HR	95%CI	*p*-Value
**Disease specific survival**	Let 7b-5p	High	Ref		
		Low	2.19	1.25–3.85	< 0.01
	Age	Per yr increase	1.01	0.99–1.03	0.3
	Stage	I	Ref		
		II	1.97	0.64–6.07	0.2
		III	6.61	2.29–19.05	< 0.01
		IV	41.62	13.49–128.41	< 0.01
**Overall Survival**	Let 7b-5p	High	Ref		
		Low	1.91	1.26–2.87	< 0.01
	Age	Per yr increase	1.03	1.02–1.05	< 0.01
	Stage	I	Ref		
		II	1.86	0.97–3.56	0.06
		III	4.05	2.15–7.63	< 0.01
		IV	15.59	7.27–33.45	< 0.01

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
