# Peer review of "Integrative Analyses of Multilevel Omics Reveal Preneoplastic Breast to Possess a Molecular Landscape That is Globally Shared with Invasive Basal-Like Breast Cancer"

_cancers, 2020, doi:10.3390/cancers12030722_

Round 1

Reviewer 1 Report

Dear Editors,

thank you very much for sending me the revised version of the paper by Zhenlin Ju , Anjana Bhardwaj.

In this revised version, the authors addressed all my concerns well.

Overall, this is an interesting study, reported and integrated investigation of the microRNA-regulatory network in a model of cancer progression. Focused on the importance of the interplay between DICER and Let-7, the paper is certainly of interest to those who work in the field of breast cancer.

Reviewer 2 Report

Dear Editor,

I have carefully checked the revised manuscript.

I am satisfied with their answers for my concerns.

I think that this revised manuscript is suitable for publication of Cancers.

Minor point

In Materials and Methods section, L 659; 570nM is a mistake of 570nm.

This manuscript is a resubmission of an earlier submission. The following is a list of the peer review reports and author responses from that submission.

Round 1

Reviewer 1 Report

No comments 

Reviewer 2 Report

In this paper, Zhenlin Ju, Anjana Bhardwaj and colleagues report an integrated bioinformatics and experimental analysis aimed at the identification of the molecular signatures operating in the first stages of neoplastic formation, in the context of MCF10A cells as model for breast cancer onset and progression.

The authors first performed a bioinformatics analysis of miRNA-seq and mRNA-seq data on a model composed by a set of breast cancer related cell lines (MCF10A-based, namely: MCFA10.P, MCF10.AT1, MCF10.DCIS, MCF10.CA1D) resampling the tumor development spectrum from early onset to late progression. From the miRNA-seq data, they first devised a set of eight expression patterns, by means of K-means clustering, in order to identify miRNAs that change significantly across the steps of the histologic progression model of breast cancer. They found a group of 213 miRNAs modulated in the eight expression patterns, most of them upregulated and most of them noted in the initial progression from normal to neoplastic cells (see Fig. 1 and Fig.2). In the second part of the work the authors focus their attention on the study of DICER and the let-7b-5p miRNA and their mutual interactions, eventually suggesting that the let-7b-5p-mediated DICER regulation causes the early global miRNA upregulation observed in the previous point (Fig. 3 and 4). In an intermediated analysis step, low levels of let-7b-5p were found to predict poor prognosis in breast cancer patients, by means of TCGA data (Fig. 5). After that, in the third part of the paper, the authors focused their attention in the characterization of the expression changes of protein coding genes. Following a mRNA-seq experiment they found that the majority of genes were deregulated in in the earliest transition from normal to preneoplastic cells and then they devised an approach to connect de-regulated miRNA-mRNA pairs (Fig. 6) based on inverse pattern of expression. Finally, the last paragraphs report the results of a procedure aimed at the elucidation of the functional pathways associated with the deregulated genes, with Fig. 7 and 8 showing a network representation of for the top two pathways derived from the transitions of normal MCF10A.P to preneoplastic MCF10.AT1 cells and MCF10.AT1 to MCF10.DCIS cells.

This paper is basically a follow-up of a previous work by the some of the authors published in 2017 (Ref. 17, Bhardwaj et al, Oncotarget 2017), where the authors originally performed the miRNA-seq and mRNA-seq experiments used here. Bhardwaj et al, 2017 focused on the in-depth study of miR-29c and its downstream pathways, suggesting, as future development, the possibility of additional investigation on additional miRNAs.

The present work is therefore the continuation of the same research line. Most of the work presented here is a computational analysis of the miRNA-seq and mRNA-seq originally performed in  Bhardwaj et al, 2017, aiming at a better definition of the reciprocal miRNA-mRNA interactions and suggesting a prominent role for the let-7b-5p – DICER crosstalk.

I have the following comments:

1)             The sequencing data presented here are already published and the current version of the manuscript is designed primarily as a second-level bioinformatics analysis. The results reported here, suggesting that most of the genomic changes in the progression to basal-like breast cancer occur in the very earliest stages of histologic progression, are important but they require, in my opinion, some additional validations to gain definitive importance. For example, the authors here provide a set of putative core pathways / genes associated to the early onset of breast cancer as final point. I suggest the authors to select one or two of the genes reported in Fig. 8 and perform investigations aimed at defining some of them as real biologically associated central mediators of the tumor progression, like, i.e. loss of function or gain of function assays in order to show their central role in the context of triple negative breast cancer.

2)             The connection between let-7b-5p and DICER is clear, but my I’m not convinced this is the only regulatory interaction able to explain the global deregulation of genes observed by the authors. In other words: if you use both Targetscan or databases of experimentally-based miRNA-gene pairs, like miRTarBase, the DICER1 gene is predicted to be target of several other miRNAs in addition to let-7b-5p. Are these miRNAs completely depleted in the biological model used by authors or are they expressed and even more, modulated ? For example, I would like to see and heatmap reporting the expression level of miRNAs targeting DICER1 extracted from Targetscan and miRTarBase. Moreover, I would add a validation like the reported in Fig. 4 also for one additional miRNAs. I also would like to see an heatmap of your data reporting the expression pattern of known Transcription Factors.

3)             The analysis based on GSEA is unclear to me. The authors report the usage of basically the entire MsigDB database, having at the end 691 target pathways. This sound strange since hundreds of pathways probability means several thousands of genes, if I understand correctly. Finally, the authors used MODULE_83 and REACTOME_ TRANSLATION as the only discussed set of genes. Could the authors explain this choice for the discussion ?

4)             There is a general problem of quality in the figures. The text in the figures is often difficult to read even with strong magnification; some plots do not show a title; in all the heatmaps the color code for cell line names is not explained (green / cyan / orange / pink). The networks in Fig. 8 are quite obscure to interpret. Could the authors provide a text file with the network written in explicit form in order to load it in graphical software like Cytoscape (i.e. the complete set of edges composing the network).

Reviewer 3 Report

The authors have demonstrated that the existence of a let-7-mediated autoregulatory negative feedback loop to be involved with early, sustained loss of let-7b-5p during breast tumorigenesis, which causes an increase in DICER mRNA and protein, eventually resulting in a global upregulation in miRNA. Although the conclusions drawn using the MCF10A-based breast cancer progression model, they found that low levels of let-7b-5p to predict poor prognosis in breast cancer patients using cancer genome data sets. While much of the genomic analysis of breast cancer is focused on characterization of invasive carcinoma and DCIS, the genomic alterations earlier in the progression spectrum has remained largely unexamined. In this way, the results are very interest because they infer that the majority of miRNA and gene alterations that defined the DCIS to invasive transition are already present in the initial transition of normal, untransformed cells to preneoplastic cells in breast cancer. I think that the manuscript is suitable for publication for Cancers with minor revision.

Comments are as follows:

Major concerns;
1) In Fig. 3A, are there significant difference for DICER mRNA between MCF10 DCIS and others?

2) In Fig. 5, authors showed that patients with low levels of let-7b-5p indicated poor prognosis in breast cancer by Kaplan-Meier analyses. The Kaplan-Meier analyses are good but not enough for their conclusion. They should also evaluate their results by multivariate Cox regression model. Multivariate Cox regression model should be used to evaluate the influence of gene expression and to estimate adjusted hazard ratios by age as a confounding factor. If possible, I also recommend for stage or/and grade as other confounding factor.

3) In Fig.5, authors focused on invasive basal-like type breast cancer in present study. Therefore, they should examine the prognosis about low levels of let-7b-5p in basal-like type breast cancer.
4) In Fig.3, the amount of dicer mRNA and protein were higher in MCF10A CA1D than MCF10A P. On the other hands, Let-7b-5p level was equally to MCF10A P in Fig. 4A. Please explain for the reason in discussion section.
5) In Fig. 4A, Let-7b-5p level was higher in MCF10 CA1D than in MCF10 AT1 and MCF10 DCIS, and equally to MCF10A P. However, their Kaplan-Meier analyses showed that patients with low Let-7b-5p level indicated poor prognosis in invasive breast cancer. Although these results were confused and controversial, it is very important point for their conclusion. They should discuss in more detail in discussion section.

Minor points

1) In Material and Methods section, there is no information for the cell numbers. Please write the number of cells used in each experiment.

2) In Material and Methods section, there is no information for methods and antibodies in immuno blotting. Please write the methods and antibodies for immunoblotting.

3) In figure 3 and 4, there is no information about statistical test. ANOVA? Please describe the information in Figure legends and/or Material and Methods section.

4) G8 was wrote in Fig 1A but missed in Fig. 1C and explanation in the sentence.

5) Please explain the meaning about red and green lines or show vertical axis in Figure 1A and 1B.

6) In Figure2, where is Let-7b-5p? Please mark it.

7) About the results in Figure2, miR-103/107 and miR-192 are also target to dicer (ref.15). Please write about them and discuss this point in discussion section.

8) Please describe the introduction about Let-7b-5p in introduction section or result section. Because the description of Let-7b-5p appears suddenly in the sentence, I confused.